# Teacher acceptability of physically active learning in UK secondary schools – a mixed methods study

**Lara E. Hollander** *, **Zoi Toumpakari, Lydia Emm-Collison**

Exercise, Nutrition and Health Sciences, School for Policy Studies, University of Bristol, United Kingdom

* lara.hollander@bristol.ac.uk

## Abstract

### Background

Roughly 70% of UK adolescents are insufficiently physically active, while secondary education reform is needed to improve adolescent wellbeing and 21st-century skill-building. One potential strategy to help address these areas is physically active learning (PAL), where movement is integrated into curricular lessons. In the UK, studies have largely focussed on primary schools; little is known about PAL in UK secondary schools. This study aimed to explore, using mixed-methods, UK secondary school teachers' acceptability of PAL, and their delivery preferences, perceived implementation barriers and facilitators.

### Methods

Cross-sectional data were collected from UK secondary teachers via online survey (N = 75). In addition to descriptive statistics, Mann-Whitney $U$ tests were conducted to examine differences by gender and school types (urban/rural, state-funded/independent), and Kruskal-Wallis tests for differences by subject. Qualitative data on teachers' current teaching practice, PAL acceptability, and perceived barriers and facilitators, collected through open survey questions (N = 63), and online teacher interviews (N = 7), were analysed using the framework method.

### Results

Using a scale of 1.0 (strong disagreement) to 5.0 (strong agreement), participants liked the concept of PAL (median 4.0, IQR 4.0,5.0) and would use PAL if it was school policy (median 4.0, IQR 3.5,5.0). Female participants were more certain that PAL should be implemented than male ($U$ = 361.5, $p$ = 0.04). Languages teachers found PAL appropriate for their subject more than humanities and social science teachers ($U$ = 6, $p$ = 0.01). Four main themes were generated: (1) 'It's time for a change'; (2)

**Data availability statement:** All relevant data are within the paper and its Supporting Information files.

**Funding:** The author(s) received no specific funding for this work.

**Competing interests:** The authors have declared that no competing interests exist.

'PAL seems like common sense'; (3) 'Is PAL realistic in secondary schools?' and (4) 'Recommendations for implementation', highlighting a collaborative approach.

## Conclusions

UK secondary teachers largely felt that PAL is a logical and enjoyable tool to contribute to education system improvement and can benefit pupils' wellbeing and learning, if appropriately executed. The findings can inform future research working towards sustainable PAL implementation in UK schools.

## Introduction

Physical activity (PA) levels decline throughout adolescence, with evidence suggesting that PA time may be replaced with sedentary behaviour [1–3]. Adolescence is a critical stage in lifelong behaviour pattern formation [4]. The UK government recommends that 11–18 year olds engage in an average of one hour of moderate-to-vigorous PA daily, of which 30 minutes should occur within schools [5]. Adolescents should also minimise sedentary time, with sedentary periods interrupted with at least light PA [5,6]. However, approximately 70% of British adolescents do not meet these recommendations [7]. Physical activity has been shown to benefit adolescent mental, cardiometabolic and bone health, while low levels of activity have been identified as a risk factor for multiple chronic diseases such as diabetes and heart disease, and premature mortality [8–13].

PA interventions delivered within the school setting are suggested to increase the equality of opportunity for young people's PA engagement [14,15], but current efforts to increase PA across the school day have not been effective [16]. One challenge has been that academic attainment tends to be prioritised over health promotion in the context of time constraints and curriculum pressures [17].

Having an overloaded curriculum has been criticised, particularly, in the UK, for encouraging narrow pedagogical styles and content, with insufficient attention given to pupil wellbeing and skills such as collaboration and creativity that could help prepare learners for the global challenges of the 21st century [18–21]. Boosting these aspects may additionally enhance enjoyment within the learning process. Enjoyment has been associated with self-determined motivation and lesson engagement; in turn promoting meaningful learning [22–24]. Enjoyment has further been identified by the Department for Education as a goal in itself for certain subjects [25]. Yet, pupils in secondary schools are reportedly the least likely of all school ages in the UK to say they enjoy school (44%) [26].

A potential strategy to help improve adolescent PA and learning within schools in an enjoyable manner and without detracting from curriculum time, is physically active learning (PAL) [27]. PAL refers to PA integrated into, or combined with, academic content during lessons outside of physical education [28]. This is separate from active breaks, where movement is unconnected to learning [29]. In PAL, movement is directly integrated with curriculum content (e.g., pupils acting out particles to illustrate

osmosis) or indirectly combined (e.g., pupils running to different corners to answer questions) [28]. PAL challenges the traditional, primarily didactic and sedentary, pedagogical approach in schools, acknowledging the neuroscientific evidence connecting PA, cognition, and learning [30]. A growing evidence base indicates that PAL improves PA, academic, enjoyment and behaviour outcomes for primary school pupils [31,32].

While PAL research has progressed at the primary school level, equivalent research at the secondary school level (ages 11–18) is lacking, despite potential similar benefits for adolescents [27,31,33,34]. Long-term adherence has been a key challenge in PAL implementation, although this reflects difficulties in sustaining school health interventions more generally [17,35,36]. Secondary schools are a different context to primary schools, with an older population, more complex curriculum content and different subjects through the day [27]. Most available PAL studies at the secondary school level have been conducted in Scandinavia [28,36–40]. The studies suggest PAL may be an effective tool for increasing PA outcomes in adolescents, and provide insight into teachers' preferences for implementing PAL, such as needing flexibility in PAL delivery to manage complexities in their teaching practice, and a whole-school approach [36,37].

However, the findings cannot be generalised to UK schools due to the impact of cultural differences in the use and context of PAL [41], e.g., the Nordic-specific concept of 'friluftsliv' (i.e., outdoor life with connection to nature) influencing the school curriculum [42,43]. To explore the potential of PAL and enhance future delivery effectiveness in UK secondary schools, there is a need for a greater understanding of how best to implement PAL-related interventions [34,44].

As the implementers, teacher buy-in and input is necessary before progressing further in adolescent PAL research, for example engaging other key stakeholders and designing intervention studies [35]. Therefore this study, aligning with the 'Feasibility' component of the UK Medical Research Council's Framework for Developing and Evaluating Complex Interventions [45], sought to examine the acceptability and feasibility of PAL amongst secondary school teachers. A mixed-methods approach was utilised to gain a rich and comprehensive understanding of UK secondary teachers' perspectives regarding hypothetical PAL implementation, specifically answering:

1) How acceptable is PAL for UK secondary school teachers across a range of subjects?

2) What are their preferences for PAL delivery?

3) What are the perceived barriers and facilitators for PAL implementation?

## Methods

### Study design

A concurrent mixed-methods research design was used to enable a full and rich understanding of different aspects of teachers' acceptability of PAL, which is imperative at this preliminary stage of adolescent PAL research [27,45–47]. The convergence of equally-weighted qualitative and quantitative elements provided a form of triangulation, allowing new insights, and the identification of inconsistencies or corroboration in results from the two methods, thereby enhancing credibility [48]. Further, results derived from qualitative data illustrated and contextualised quantitative findings [46,48]. The decision to employ a concurrent rather than sequential approach, and use a cross-sectional design, was taken to ensure sufficient data collection could be conducted within the available time frame. Quantitative and qualitative data were collected and analysed in parallel, after which the respective results were compared to identify patterns that support each other or diverge, and how the qualitative findings might explain quantitative ones. Key connections were then narratively integrated into the discussion section.

Ethical approval was obtained from the School for Policy Studies Research Ethics Committee at the University of Bristol on 07/06/2022 (Reference: 11459).

## Participants and recruitment

Teachers working in UK secondary schools or colleges, teaching pupils aged 11–18 years old, were eligible to participate. Teachers of any subject or in the senior leadership team were permitted, and schools could be state-funded or independent, facilitating a broad understanding of the secondary school landscape for PAL. As this is an exploratory study, the recruitment target was 75–150 survey responders, with a subsample of 7–10 interviewees, based on previous similar studies [38,49–53]. Due to the recruitment of participants largely through social media groups, and personal and professional contacts, it was not possible to track the number of individual schools represented in the sample. As such, no minimum school count was set. Instead, specific characteristics were looked for to enable diverse perspectives to be captured, such as teachers in different school types (rural, urban, state-funded, private, special school), teachers of a variety of subjects, and the representation of heads of year and senior leadership team. Thus, some recruitment was purposive. Feedback from secondary teachers and staff during recruitment highlighted time pressures for this population. Therefore, to gain sufficient participant numbers, convenience sampling was also used when needed, and some snowball sampling.

While interview and survey recruitment were combined, an option at the end of the survey also provided space for participants to leave their email address if interested in participating in an interview. Teachers who indicated interest were emailed with further information and a convenient time was arranged. The recruitment process is outlined in Fig 1.

Purposive sampling was undertaken through social media recruitment in addition to emails and phone calls to school gatekeepers. Social media recruitment is considered efficient [54] and the presence of education hashtags on Twitter and

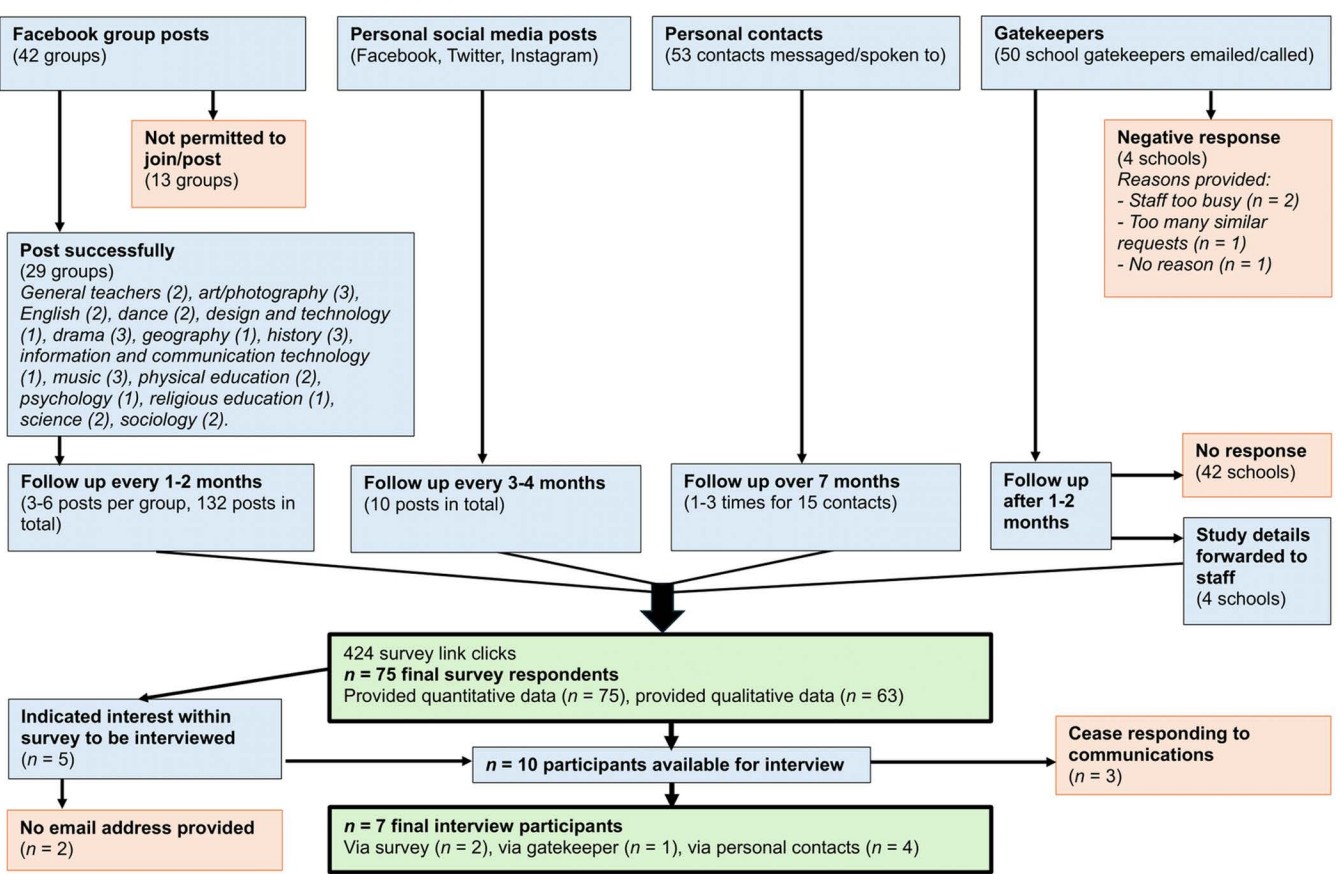

**Fig 1. Recruitment process between July 1st 2022 and February 17th 2023.**

subject-specific Facebook groups were beneficial to accessing relevant teachers of particular subjects. Contacting school gatekeepers, usually school receptionists, allowed the targeting of specific locations (e.g., urban and rural) and types of school (e.g., state-funded and independent). The information emailed to gatekeepers included a brief study background, aims and details about the survey and interviews.

Study information sheets were included at the start of the online survey, after which participants could digitally confirm their informed consent. Separate information sheets were emailed to potential interviewees, and a consent form which participants digitally signed and returned.

## Data collection

### Survey

Quantitative and qualitative data were collected through an anonymous online survey hosted on Jisc [55]. Participants first provided demographic and professional background information, including age, gender, ethnicity, disability status, parental status, employment pattern (full-time or part-time), school region (within UK), setting (urban or rural), gender composition (single or mixed), school type (state-funded or independent), subjects and year groups taught, average class size, and any additional roles held.

Participants were first asked about their current classroom movement practices. A short video (~12 minutes) presented an overview of PAL [56], after which participants indicated whether they had watched all, some, or none of it. Participants were then asked about their reactions to PAL (including its perceived value and how they expect pupils would respond), preferred PAL delivery formats, and the perceived feasibility of its delivery in their context. Open-ended questions were included throughout to allow participants to elaborate or share broader reflections, for example noting one aspect they liked about PAL, one they did not, and any further comments.

The survey was informed by the PAL literature [34,35,38–41,57,58]. Most survey items were newly developed by the researcher and used a five-point Likert scale. However, two measures were adapted from existing tools [51,59]. Initial reactions to PAL were assessed using a series of positively and negatively-framed statements, adapted from a tool used by Emonson et al. [51]. Motivation to use PAL was measured using an adapted version of the Work Tasks Motivation Scale for Teachers [59]. This tool has shown, amongst Canadian secondary school teachers, convergent and discriminant validity, and acceptable levels of internal consistency, with Cronbach's alpha coefficients of <.70 [59]. Irrelevant items were removed, the wording adjusted to suit the PAL context (e.g., "I would feel bad if I didn't start to use PAL sometimes"), and the response scale reduced from seven to five points for consistency.

Teachers were also asked whether they would consider including movement in future lessons and if they would be likely to adopt PAL given appropriate training and support, with Yes, No, or Maybe response options. Question stems and response scales can be found in the Results section and S1-S4 Tables.

The full survey was piloted with one secondary school teacher and three peers prior to distribution, to establish face validity [60]. They provided feedback on the survey's clarity, length, and functionality. The teacher also commented on the usefulness of the video and the appropriateness of the language for the target population. The survey was subsequently revised before formal launch.

### Interviews

A sub-sample of seven secondary school teachers, recruited through personal contacts, a school gatekeeper, and the survey, participated in one-to-one interviews, conducted online via Zoom to facilitate joining from home or school at their convenience. Participants completed the survey, and were asked to watch or rewatch the TEDx Talk video [56] prior to the interview. Interviews lasted 30–45 minutes to allow participation within a free period. Interviews and analysis were conducted by LEH (MSc), a female student with MSc level training in qualitative and quantitative methods and experience of

 

qualitative research within schools. Reflexivity was maintained through the use of a reflexive diary and regular supervisory discussions, helping to identify potential biases and support transparency throughout the research process.

The interactive and partially flexible format of semi-structured interviews elicited in-depth responses from participants and permitted emergent topics to be explored, while sufficient structure remained to capture relevant data with comparability across interviews [61]. The first interview served as a pilot interview, with participant feedback and the interviewer's own reflections informing subsequent interviews. An interview topic guide can be found in S1 File.

To begin, participants were reminded about their data confidentiality, were asked to verbally reconfirm consent and were given the opportunity to ask any questions. The interview started with an ice breaker and subsequently explored, in more detail, teachers' current in-class movement practices, their acceptability of PAL, and perceived barriers and facilitators. Some questions were adapted from existing studies to be specific to PAL (e.g., 'Can you imagine integrating this program in your teaching practice?' to 'Can you imagine integrating PAL into your teaching practice?') [51,62]. Topic guides were tailored to teachers of active versus non-active subjects and to those who held additional roles such as Head of Department. Data provided were summarised and checked with participants where possible during the interview, to ensure responses were being understood as intended.

## Quantitative analysis

Survey-derived quantitative data were analysed using IBM SPSS 29.0 [63]. The data were screened for outliers and frequency analysis identified any missing data, which were excluded pairwise in analysis to maximise available data [64].

Descriptive statistics were reported as raw frequencies and percentages, median, and interquartile range (IQR). Variables included demographics and aspects of PAL acceptability and delivery. Tests of difference for PAL acceptability and delivery were conducted; two-tailed Mann-Whitney $U$ tests for differences by teachers' 1) gender 2) school type (rural or urban) and 3) school type (state-funded or independent), and Kruskal-Wallis tests for differences by subject category (arts, languages, maths and sciences, humanities and social sciences, physically active subjects).

## Qualitative analysis

The framework method was used to analyse interview and survey-derived (open-question) qualitative data. This approach is flexible and systematic, appropriate where the data covers analogous topics, and often used in the thematic analysis of semi-structured interview transcripts [65]. It facilitated thick descriptions of phenomena to benefit transferability of the findings, aided by the complexities inherent in the layered matrix structure where context is maintained for each summarised piece of data [65,66]. NVivo R1 (2020) [67] was used to facilitate analysis.

Reporting was guided by the Consolidated Criteria for Reporting Qualitative Research [68] and analysis conducted by LEH. The analysis procedure comprised seven stages [65]:

1) Verbatim transcription – Interviews were audio-recorded and transcribed verbatim through Zoom followed by manual checking for accuracy and anonymising. Reflexive and initial thoughts were written by LEH before, during, and following each interview.

2) Familiarisation with the data, supported by reflexive and analytical notes – Familiarisation by listening to interview audio recordings and reading transcripts.

3) Coding of interview transcripts and survey qualitative responses – This Initial inductive coding, both semantic and latent, allowed data-driven and unexpected insight.

4) Developing a working analytical framework – It soon became evident that many codes aligned with the socio-ecological model, reflecting previous related research [57,62,69–71]. Thus, the analytical framework, combining data from both interview and open survey questions, became largely organised by this model (see S2 File). It is important to note

 

that while the socio-ecological model was used to help structure analysis, the final themes were not mapped onto this model.

5) Applying the analytical framework, including a second round of coding in reverse order of participants [72].

6) Charting summarised data into the matrix, still largely structured by the socio-ecological model, retaining participants' meaning and language wherever possible

7) Data interpretation – Through examining the framework matrix cells while noting and reflecting on connections and prominent notions. Potential themes and subthemes were drafted and refined as the interpretation progressed.

By the end of stage seven, final themes and their suggested relationships were identified. The qualitative findings were used to add context to and explain the quantitative findings.

## Results

Table 1 provides an overview of participant demographics across the survey and interview.

### Quantitative analysis

Data from 75 participants were included in quantitative analysis. For context, the majority were female and white Caucasian (Table 1), and non-disabled (97.3%). Most participants worked full time (70.7%) in state-funded, urban schools (Table 1) that were mixed-gender (82.7%). Year groups taught were spread fairly evenly between Years 7–13, and most class sizes comprised ≤26 pupils (50.7%).

### Current class PA

Fig 2 summarises the findings examining teachers' current class PA. On average, participants provided some movement opportunities within classes (median 3, IQR 2,5) and strongly agreed that students should move more through the school day (median 5, IQR 4,5). Female teachers provided more opportunities than male teachers ($p = 0.02$). Due to n = 1 transgender teacher, they could not be included in gender comparisons in analysis. Teachers at rural schools were more likely to disagree that students should stay sat down in classes than those at urban schools ($p = 0.03$). No significant differences were found between state and independent schools.

### PAL acceptability

As seen in Fig 3, most participants liked the idea of PAL (median 4, IQR 4,5). Participants were willing to participate in training if offered and agreed that PAL should be implemented in secondary schools (all median 4, IQR 3,5). Compared to female teachers, male teachers were less sure that PAL should be implemented ($p = 0.04$), though both groups would be likely to incorporate PAL into lessons with appropriate training and support (median 3, IQR 2,3). More teachers at rural schools believed that PAL would improve pupils' focus and behaviour than teachers at urban schools ($p = 0.047$). No significant differences were found between state and independent schools.

Fig 4 illustrates how effective participants believed PAL would be for different year groups.

Regarding the gender of pupils that teachers thought would be most receptive to PAL, 49.3% selected boys, 2.7% selected girls, and 47.9% felt both would be equally receptive.

### PAL delivery

Fig 5 shows the key results for measures investigating PAL delivery. Amongst teachers of physically active subjects, 70.7% would collaborate with other subjects to develop and deliver PAL. Teachers at urban schools were more likely to expect that having adequate space would impact PAL adoption than rural schools ($p = 0.04$), while teachers of

**Table 1. Summary of study population characteristics used in analysis.**

| Characteristics | Survey (quantitative) Frequency (%) *N*=75 | Survey (qualitative)* Frequency (%) *N*=63 | Interview* Frequency (%) *N*=7 |
|---|---|---|---|
| **Gender** | | | |
| Female | 55 (73.3) | 49 (77.8) | 4 (57.1) |
| Male | 19 (25.3) | 13 (20.6) | 3 (42.9) |
| Transgender | 1 (1.3) | 1 (1.6) | 0 (0.0) |
| **Age in years** | | | |
| 18-24 | 1 (1.3) | 1 (1.6) | 0 (0.0) |
| 25-34 | 25 (33.3) | 23 (36.5) | 1 (14.3) |
| 35-44 | 28 (37.3) | 21 (33.3) | 4 (57.1) |
| 45-54 | 12 (16.0) | 10 (15.9) | 0 (0.0) |
| 55+ | 9 (12.0) | 7 (11.1) | 2 (28.6) |
| **Ethnicity** | | | |
| Asian/Asian British | 3 (4.0) | 3 (4.8) | 0 (0.0) |
| Black/African Caribbean/Black British | 1 (1.3) | 1 (1.6) | 0 (0.0) |
| White/Caucasian | 69 (92.0) | 56 (88.9) | 7 (100.0) |
| Other | 1 (1.3) | 1 (1.6) | 0 (0.0) |
| Missing | 1 (1.3) | 1 (1.6) | 0 (0.0) |
| **School location** | | | |
| Rural | 21 (28.0) | 17 (27.0) | 4 (57.1) |
| Urban | 54 (72.0) | 46 (73.0) | 3 (42.9) |
| **School type** | | | |
| Independent | 14 (18.7.0) | 11 (17.5) | 3 (42.9)** |
| State-funded | 61 (81.3.0) | 52 (82.5) | 4 (57.1) |
| **Subject category** | | | |
| Arts | 3 (4.0) | 3 (4.8) | 0 (0.0) |
| Humanities & social sciences | 16 (21.3) | 11 (17.5) | 0 (0.0) |
| Languages | 6 (8.0) | 6 (9.2) | 2 (28.6) |
| Maths & Sciences | 17 (22.7) | 13 (20.6) | 2 (28.6) |
| Physically active classes | 32 (42.7) | 29 (46.0) | 2 (28.6) |
| Other | 1 (1.3) | 1 (1.6) | 1 (14.3) |
| **Additional role** | | | |
| Assistant headteacher | 4 (5.3) | 1 (1.6) | 0 (0.0) |
| Head of department/year/house/faculty | 28 (37.3) | 24 (38.1) | 1 (14.3) |
| Assistant head of department/year/house/faculty | 7 (9.3) | 5 (7.9) | 2 (28.6) |
| Other | 10 (13.3) | 10 (15.9) | 0 (0.0) |
| No additional role/not reported | 26 (34.7) | 27 (42.9) | 4 (57.1) |

*Note: some participants may be represented in both columns, if they completed both survey and interview **Includes 1 special school

state-funded schools were more likely to expect school administration support to impact PAL implementation compared to teachers at independent schools (*p*=0.002).

Fig 6 shows participants' preferred delivery formats of PAL. Outdoor learning was the most popular choice, and PAL was preferred over movement breaks.

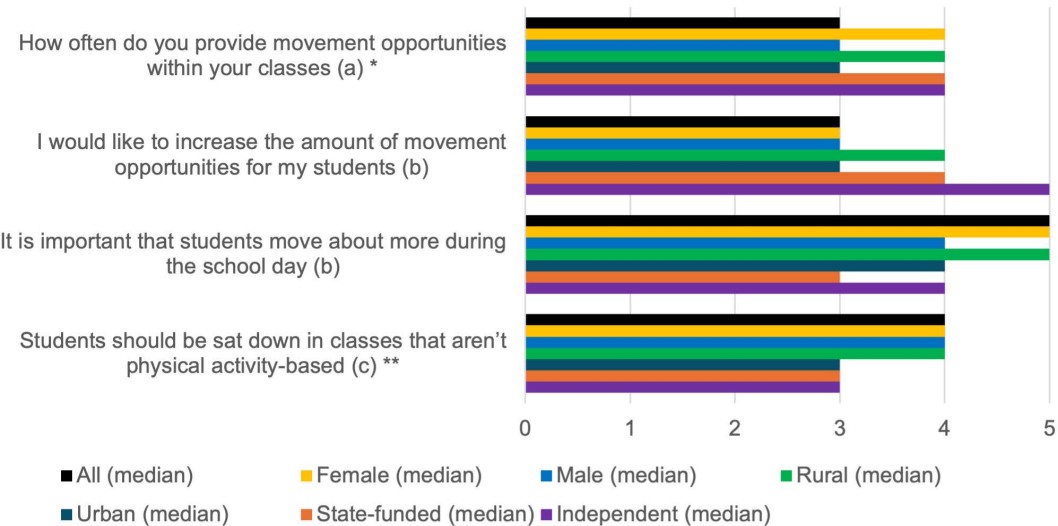

**Fig 2. Summary of results for current classroom physical activity, with differences by participant gender, school location, and school type.** * = statistical significance of p<0.05 for male vs female teachers. ** = statistical significance of p<0.05 for urban vs rural schools. (a) Scale 1–5: 1 = Never, 3 = Sometimes, 5 = Always. (b) Scale 1–5: 1 = Strongly Disagree, 3 = Don't Know, 5 = Strongly Agree. (c) Scale 1–5: 1 = Strongly Agree, 3 = Don't Know, 5 = Strongly Disagree.

## Tests of difference by subject

Kruskal-Wallis tests with post-hoc analysis were run to identify any significant differences between subject categories, for nine variables which were selected to cover key aspects of current class PA, PAL acceptability and PAL delivery (Table 2). Language teachers were significantly more likely to find PAL appropriate for their subject than humanities and social science teachers ($p = 0.01$), with the latter group tending to respond more negatively towards PAL across the variables.

## Qualitative analysis

All seven interviewees had watched the TED talk video prior to interview. Interview and qualitative data from open-ended survey questions were combined in a framework matrix to facilitate data interpretation and the identification of themes.

Four central themes (Fig 7) were produced relating to PAL acceptability and participants' thoughts about hypothetical delivery: 1) It's time for a change; 2) PAL seems like common sense; 3) Is PAL delivery realistic in secondary schools?; 4) Recommendations for implementation.

To elaborate on the proposed relationships between themes, participants felt PAL could be an acceptable and logical option (Theme 2) to help generate change in an education system that was considered to need new approaches (Theme 1). Barriers to PAL delivery described by teachers (Theme 3) may stem from perceived flaws in the education system (Theme 1). Equally, the concerns conveyed by participants (Theme 3) further fuel the conveyed notion that the education system more widely requires change (Theme 1). Implementation recommendations provided by teachers (Theme 4), could build on existing practice and facilitate the perceived benefits for pupils (Theme 2), potentially overcoming some apprehensions (Theme 3). Examples of teacher responses to illustrate these themes are presented in S3 File. Some additional quotes are also presented in the text.

### Theme 1: It's time for a change

This theme illustrates a conveyed sense that change is needed within the wider education system and policy, for the benefit of young people's learning, health, and wellbeing.

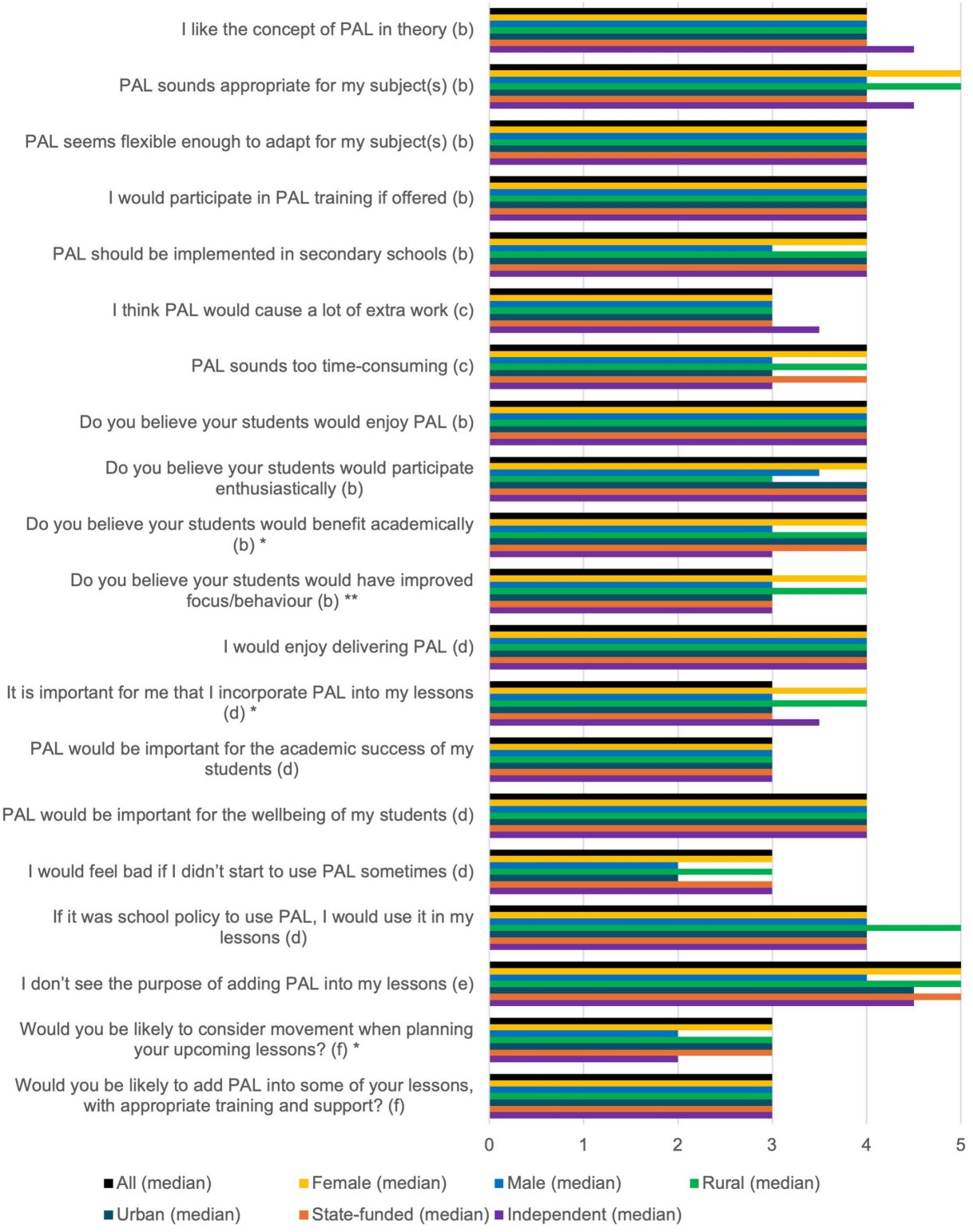

**Fig 3. Summary of results for PAL acceptability, with differences by participant gender, school location, and school type.** * = statistical significance of p<0.05 for male vs female teachers. ** = statistical significance of p<0.05 for urban vs rural schools. (b) Scale 1–5: 1 = Strongly Disagree, 3 = Don't Know, 5 = Strongly Agree. (c) Scale 1–5: 1 = Strongly Agree, 3 = Don't Know, 5 = Strongly Disagree. (d) Scale 1–5: 1 = Does not correspond at all

with how I feel, 5 = Corresponds completely with how I feel. (e) Scale 1–5: 1 = Corresponds completely with how I feel, 5 = Does not correspond at all with how I feel. (f) Scale 1–3: 1 = No, 2 = Maybe, 3 = Yes.

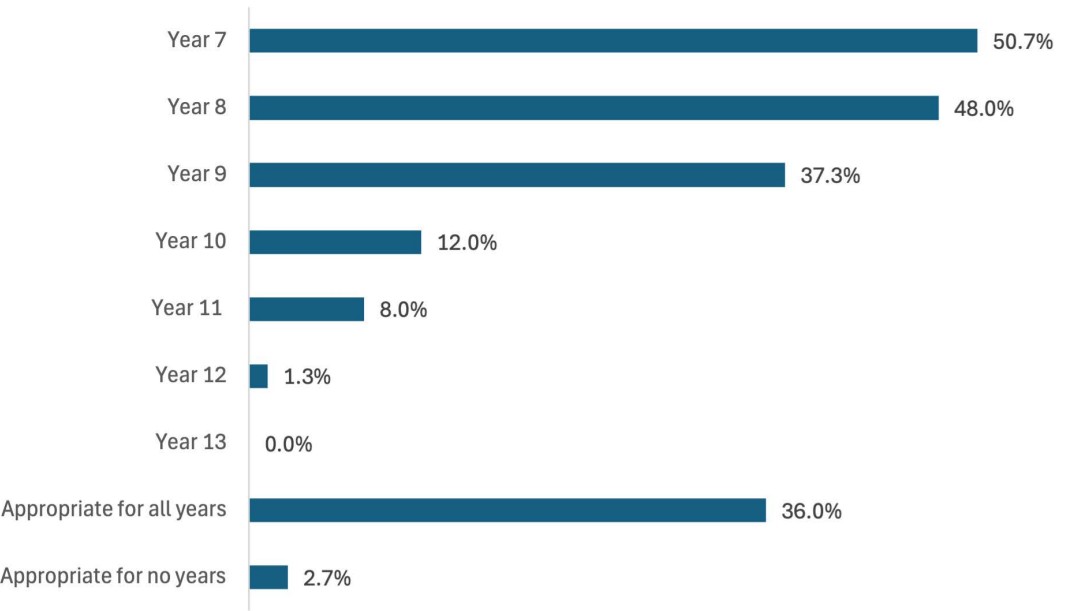

**Fig 4. A bar chart of participant responses to 'Do you think that PAL would be most effective for certain year groups?'.** (<100% across options since multiple selections allowed).

### We need to move on from a Victorian view of education

Participants perceived limitations to the education system and 'traditional learning' approaches. They suggested the current approach is outdated, which they observed within their schools and classrooms as involving a prioritisation of more traditionally academic subjects and written examinations, with pupils often sat quietly for long periods and engaging in rote learning. Teachers perceived a necessity to update the approach, to better reflect the contemporary world. Within this, teachers indicated a crucial education system flaw was insufficiently accommodating difference amongst pupils, whether due to special education needs, some pupils needing more movement, or general learning style variances. For example, one participant described how there may be different types of pupils, some that learn better sitting at a desk and some who don't, which can mean for the latter group, *"you're sort of slightly failing those ones" (P3, female, Special School).*

### National policies have exacerbated the neglect of movement in schools

Participants described how 2016 education reforms made examinations much more demanding to prepare for, consequently deprioritising more physically active subjects and activities in schools. Residual behaviours from COVID-19 restrictions were suggested to have further reduced opportunities for movement. Teachers consequently felt that PA needed to be returned into focus within schools.

### Theme 2: PAL seems like common sense

This theme illustrates how teachers felt PAL could be a suitable tool to contribute to improving the education system, and perceived the concept of PAL as logical.

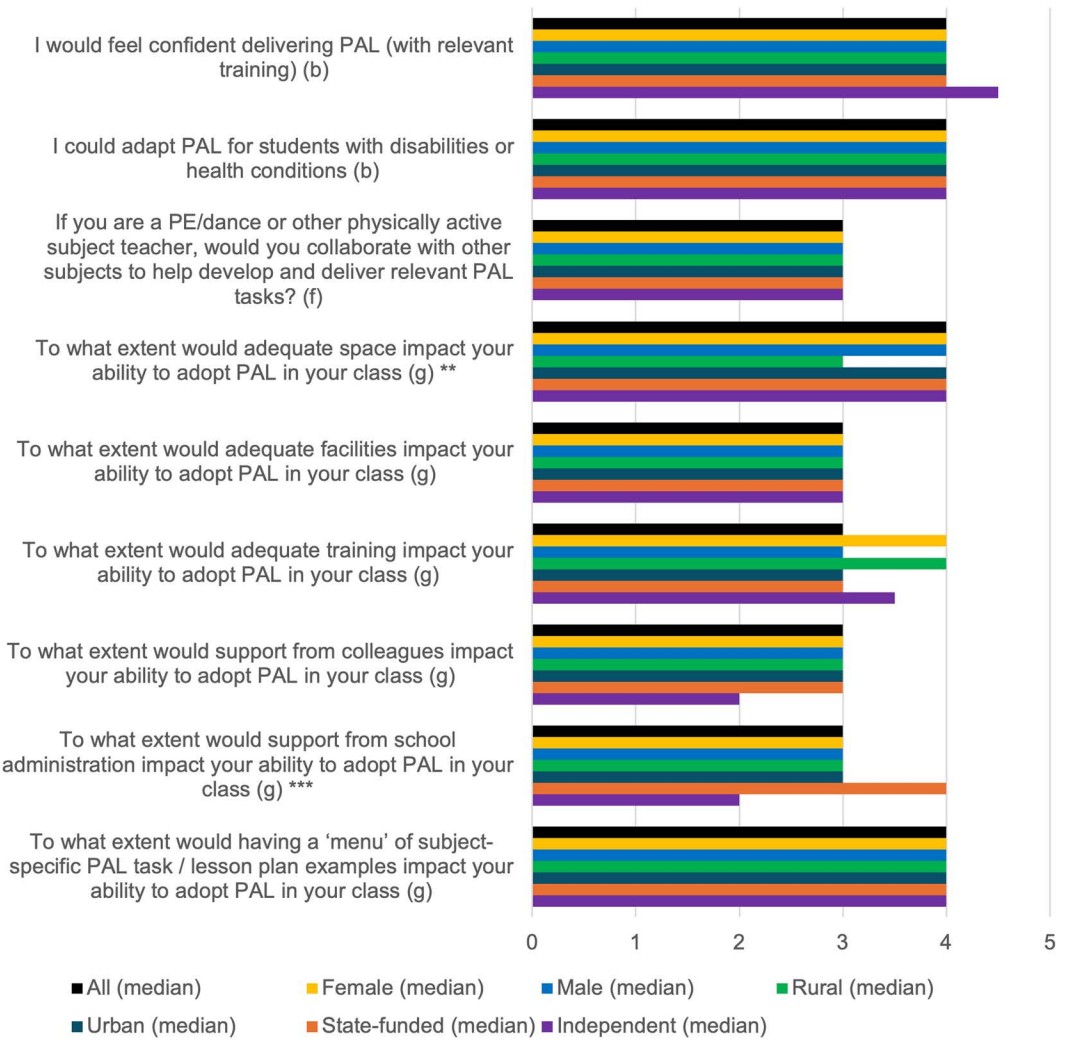

**Fig 5. Summary of results for PAL delivery, with differences by gender, school location, and school type.** ** = statistical significance of p<0.05 for urban vs rural schools. *** = statistical significance of p<0.05 for state vs independent schools. (b) Scale 1-5: 1=Strongly Disagree, 3=Don't Know, 5=Strongly Agree (f) Scale 1–3: 1 = No, 2 = Maybe, 3 = Yes. (g) Scale 1–5: 1 = No impact at all, 5 = Strongly impact. *Missing data from two participants in items 6–9; from three in items 4–5. N = 41 participants answered item 3 as a physically active subject teacher.*

## Some classroom movement already provided

Although participants reported that PAL was not school policy or often included in teacher training, many spoke about already providing movement opportunities within their classes, and finding it valuable. For example, language teachers described engaging pupils in translation relays, and PE and drama teachers collaborated with other subjects so pupils could physically enact topics like the respiratory system or historical battles.

## PAL is perceived to have important benefits

Participants imagined PAL would have important benefits for adolescents, including physical health, mental wellbeing, academic attainment, and transferable skills such as confidence and teamwork, that are *"… kind of obvious if you look into it" (P1, female, languages).* Teachers suggested that the enjoyment and engagement inherent in PAL tasks were key

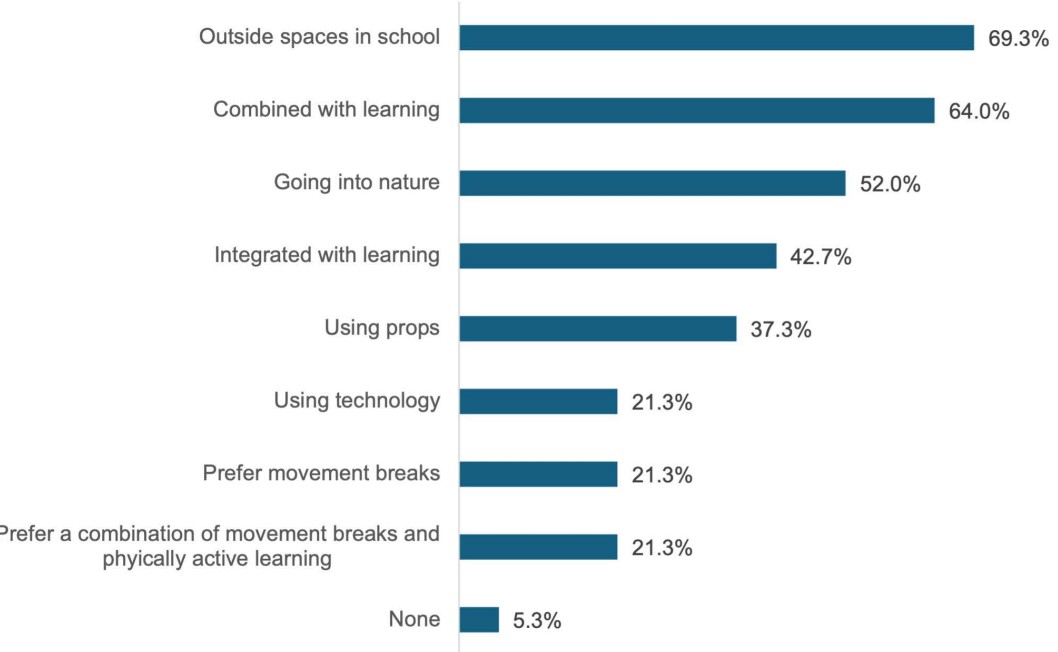

**Fig 6. Participants' preferred PAL delivery formats; amount of votes per delivery type.** (<100% across options since multiple selections allowed).

to pupils acquiring knowledge in a meaningful way. Further, teachers felt pupils' attention spans aren't large enough to sit for full lessons. Teachers valued variety within and between lessons for effective learning, which was perceived to reduce boredom and increase concentration.

> "…classroom-based lessons are more boring and I can feel when the students start to switch off because they have been sitting for too long." (P15, female, drama)

The variety PAL could offer was viewed to "…*break up monotonous learning*" (P46, transgender, dance/PE). Although, one teacher felt that it was important to encourage boredom as "… *an important life skill*", which PAL could hinder *(P14, female, sciences).*

PAL was considered a means of making education more inclusive for neurodiverse pupils, who might benefit from more physical learning. This was expressed by teachers at mainstream schools as well as special schools, for example one special school with highly autistic students who "… *learn best when flapping, moving around, and when it looks as if they're not listening but they actually are." (P3, female, special school).*

### Theme 3: Is PAL delivery realistic in secondary schools?

While teachers generally welcomed the idea of PAL, concerns were raised about the reality of delivery in the UK secondary school setting.

### Examination and curriculum pressures

Of particular concern were the pressures to get through large volumes of curriculum content in a short time, and a focus on examination outcomes. Teachers perceived they could get through curriculum content faster with the approach they're

**Table 2. Statistically significant differences in current class physical activity, PAL acceptability, and PAL delivery variables, by subject category.**

| Statement | Arts Median (IQR) n=3 | Languages Median (IQR) n=6 | Maths & sciences Median (IQR) n=17 | Humanities & social sciences Median (IQR) n=16 | Physically active subjects Median (IQR)n=32 | Kruskal-Wallis χ2 (p-value)* | Post-hoc Mann-Whitney U tests (p-value)** |
|---|---|---|---|---|---|---|---|
| **How often do you provide movement opportunities within your classes?**[a] | 3 (2,3) | 3 (2,3) | 3 (2,3) | 2 (1.25,3) | 5 (4,5) | 52.05 **(<0.001)** | Arts Vs physically active subjects: 1 **(0.01)** Languages Vs physically active subjects: 2 **(0.002)** Maths & sciences Vs physically active subjects: 33.5 **(<0.001)** Humanities & social sciences Vs physically active subjects: 2.5 **(<0.001)** |
| **I like the concept of PAL in theory**[b] | 4 (3.5,4) | 4.5 (3.75,5) | 4 (3,5) | 3 (2,4) | 5 (4.25,5) | 34.74 **(<0.001)** | Arts Vs physically active subjects: 8 **(0.04)** Maths & sciences Vs physically active subjects: 128 **(0.005)** Humanities & social Sciences Vs Physically Active Subjects: 24 **(<0.001)** |
| **PAL sounds appropriate for my subject(s)**[b] | 3 (2.5,3.5) | 4 (4,5) | 3 (2,4.5) | 2 (1,3) | 5 (5,5) | 47.7 **(<0.001)** | Arts vs physically active subjects: 2 **(0.001)** Languages vs humanities & social sciences: 6 **(0.01)** Languages vs physically active subjects: 44 **(0.03)** Maths & sciences vs physically active subjects: 80 **(<0.001)** Humanities & social sciences vs physically active subjects: 6 **(<0.001)** |
| **PAL sounds flexible enough for my subject(s)**[b] | 3 (2.5,3.5) | 4 (4,4.25) | 4 (3,4.5) | 2 (2,4) | 5 (4,5) | 32.08 **(<0.001)** | Arts vs physically active subjects: 6 **(0.04)** Maths & sciences vs physically active subjects: 122 **(0.008)** Humanities & social sciences vs physically active subjects: 43.5 **(<0.001)** |
| **I would enjoy delivering PAL**[c] | 4 (3.5,4.5) | 4 (3.75,5) | 4 (3,4.75) | 3 (1,4) | 5 (4,5) | 28.27 **(<0.001)** | Maths & sciences vs physically active subjects: 123.5 **(0.01)** Humanities & social sciences vs physically active subjects: 41.5 **(<0.001)** |
| **Would you be likely to add PAL into some of your lessons, with appropriate training and support?**[d] | 3 (3,3) | 3 (2,3) | 3 (2,3) | 2 (1,2) | 3 (3,3) | 26.78 **(<0.001)** | Humanities & social sciences vs physically active subjects: 53 **(<0.001)** |

*All df=4 **Post-hoc tests: pairwise comparisons using Mann-Whitney U test with Bonferroni correction to control for familywise error

[a] Scale 1–5: 1=Never, 3=Sometimes, 5=Always

[b] Scale 1-5: 1=Strongly Disagree, 3=Don't Know, 5=Strongly Agree

[c] Scale 1–5: 1=Does not correspond at all with how I feel, 5=Corresponds completely with how I feel

[d] Scale 1-3: 1=No, 2=Maybe, 3=Yes

Abbreviation: IQR – interquartile range. Bold p-value numbers represent statistical significance of $p < 0.05$.

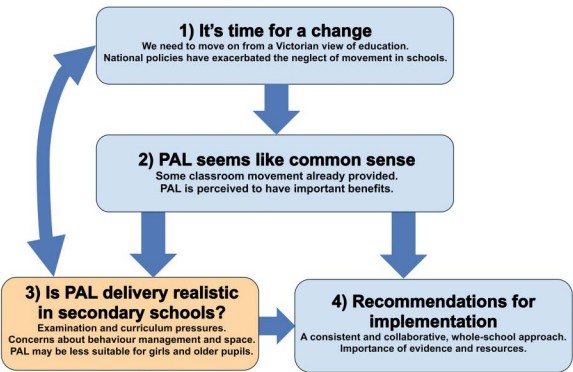

**Fig 7. Thematic map of identified themes and their suggested relationships.**

used to, with pupils sat down. They expressed concerns that staff might be reluctant to alter their teaching practice, especially if they have taught for a long time or due to recent increased pressures.

However, there also was a sense amongst some participants that PAL might not require significant extra work, particularly with suitable support.

*"I think there's always a fear, as I say there's so many pressures on teachers at the moment that you can end up thinking, "Oh, no, not another idea." You know, "How much extra work is this going to entail?" But I don't think it necessarily does have to. As long as teachers feel confident, then the preparation should be […] relatively minimal." (P6, female, languages)*

### Concerns about behaviour management and space

Teachers with suitable space available for PAL recognised that without it, PAL delivery could be limited. This perception was supported by teachers who described having less appropriate spaces. For example, subject-specific spaces like science classrooms with fixed benches and equipment, or large class numbers in small classroom spaces, due to funding issues. A particular challenge conveyed was behaviour management whilst pupils are out of their seats, particularly when in restricted spaces.

### PAL may be less suitable for girls and older pupils

Mirroring the quantitative findings, participants had observed and expect that boys need, and engage more with, class movement, while girls could *"…be more reticent." (P69, female, languages)*. Teachers noted that for girls, changes during puberty can discourage participation in physical activity and increase self-consciousness. Although a teacher described how PAL would still benefit and be received well by many girls who would be *"…more than happy to do lots of PAL given the opportunity." (P1, female, languages)*.

Assessment preparation for older pupils was considered to limit the possibility of adding PAL to lessons, due to increased curriculum content and a need, perceived by some teachers, for pupils to be used to being still for exams. It was suggested that a compromise would be having more PAL in the earlier years of secondary school. Particularly, PAL might ease their transition into an overwhelming secondary school environment, when, *"They almost need a bit more movement." (P1, female, languages).* However, one participant expressed value in PAL for older year groups, due to less physical activity outside of school and being sat down more for exam preparation, when, *"…actually that's a time in their*

*life where they should be continuing with that real passion for the subject and seeing it in different places and different contexts." (P4, female, maths).*

### Theme 4: Recommendations for implementation

The final theme highlights teachers' key suggestions for successful PAL implementation in secondary schools.

### A consistent and collaborative, whole-school approach

Participants felt a school-wide approach was needed for the most impact, facilitated by school policy and consistent integration across classes and the curriculum. Support and buy-in from the senior leadership team was considered an essential part of this, otherwise, "…*the rest of the school won't be on board as well.*" (P6, female, languages). Collaboration between staff was recommended; sharing ideas and experiences, and engaging in department collaborative planning. Cross-curricular collaboration involving PAL was welcomed by teachers, some of whom had previously experienced it. Teachers of physically active subjects were seen as a vital part of developing this form of collaboration:

> *"I'd be really keen to, you know, try and look at the idea of teaching something from a different concept, or looking at changing, you know, the way that something gets taught. […] as teachers in you know, drama, dance, and music a little bit. Um, PE […] those people are are really well placed to be able to do that."* (P5, male, drama)

### Importance of evidence and resources

Participants placed a high value on evidence-based teaching practice, and conveyed that stronger evidence for the benefits of PAL would help them accept and adopt the method. Evidence supporting PAL implementation was considered essential for persuading other key stakeholders, particularly within senior leadership teams, who tend to be results-driven. Additionally, access to resources, predominantly subject-specific PAL activity examples, was seen as vital for successful adoption.

## Discussion

A mixed-methods approach was used to gain understanding of the acceptability and hypothetical delivery of PAL for UK secondary school teachers. Findings support and extend previous PAL research in finding differences by gender, subject, and school type in PAL acceptability and implementation at the secondary school level and providing rich insight into the perspectives of UK secondary school teachers. Teachers largely found the idea of PAL to be acceptable and logical. However, participants highlighted concerns that they felt would need consideration for successful implementation, such as a pressured environment of high-stakes testing and managing pupils' behaviour.

PAL was a new concept to many teachers, although some had encountered it without this terminology. They had a good understanding of what PAL is and its rationale after watching a video, although a small minority of survey respondents appeared to confuse PAL with movement breaks or PE, which may have impacted their ability to convey their thoughts on PAL. This may be due to not having watched the video, so is perhaps indicative of their interpretation of the concept by name only. This was a concern raised in a previous study, with the name 'movement-centred pedagogy' proposed to replace PAL to enhance clarity and acceptance amongst teachers [73]. While participants' pupils mostly remained seated in classrooms, some occasions of movement within academic learning were recounted, which offers a teacher-generated base from which to build during future research and implementation, rather than a 'top-down' enforcement of PAL.

Participants generally liked the concept of PAL, felt it should be implemented, and preferred PAL to movement breaks due to the perceived purposefulness and educational benefits of integrating learning. This preference may be more

pertinent in secondary schools with the increase in examination pressures since primary school. In the context of wider education system challenges, teachers felt that change is needed and PAL could be a logical part of that. They believed it would benefit pupils and that they would enjoy delivering it. This acceptability is consistent with previous research showing that secondary teachers have a positive attitude to PAL, with the approach seen as relevant and beneficial to pupils [36,38–40,44]. In the present study, quantitative analysis indicated higher acceptability of PAL amongst female teachers compared to male, differing from a Norwegian study which observed no significant gender differences in surveyed secondary teachers' use or likelihood to enact classroom-based PA, perhaps due to cultural context [74]. This important new observation suggests further exploration is needed to assess why UK-based male teachers might be more resistant than female, and what could overcome it. One indication in the present study is that appropriate, targeted training and support may help to facilitate acceptance amongst male teachers.

Other than already physically active subjects, languages teachers were the most eager to adopt PAL. This enthusiasm is supported by a meta-analysis of school-based RCTs, which reported that PA interventions positively impacted language skills amongst children and adolescents, such as reading, vocabulary, and comprehension [75]. Further, literature on embodied cognition, a neuroscience-backed theory whereby sensorimotor experiences mediate cognitive processes, indicates that semantically linked movement enhances short- and long-term memorisation in language learning [76–80]. However, no previous research has specifically compared the suitability or acceptability of PAL across secondary school subjects. While larger-scale evidence is needed, this study provides initial insight into how PAL implementation in secondary schools might be most effective when strategically targeted within a whole-school approach, where the overarching commitment to PAL is shared school-wide, but practical integration is phased in by prioritising subjects where it is currently most acceptable and feasible. In the short term, this strategy may help build early momentum, demonstrate proof of concept, and strengthen staff buy-in. Over time, the insights gained could inform efforts to adapt PAL for subjects with lower initial acceptability, supporting a more inclusive and sustainable whole-school approach. In this way, targeted implementation and whole-school change are not competing strategies but supportive elements of a unified approach.

PAL and embodied learning research in general have tended to focus more on maths, science, and languages, rather than the humanities and social sciences [78,81–84]. Humanities and social science teachers showed the least acceptability in this study. Qualitative survey responses indicated a particular need to focus on reading and writing for exam requirements in these subjects, with limited curriculum time available for PAL. However, no humanities or social science teachers were interviewed, and therefore the perceptions of acceptability have not been explored in more detail. This should be a focus of future research, but as an initial finding raises the question of whether all subjects should be included in PAL implementation and if there are ways to overcome subject-specific barriers.

The present study's quantitative and qualitative analysis suggests PAL is viewed as potentially appropriate for all ages, but particularly the first years of secondary school. In interview, teachers spoke of the potential of PAL to aid the transition from primary to secondary school and noted that less examination pressure in the earlier years permitted more time for PAL. There has been evidence of increased sedentary behaviour and decreased PA during the transition to secondary school [85,86] and PAL could therefore offer an opportunity to minimise the negative behavioural shift during this key period of change.

Teachers expressed concern regarding classroom space, which appeared to be a more important factor for urban-located schools than rural. This is supported by previous cross-sectional research which found significant differences in pupils' self-reported time spent in PA between urban, suburban and rural secondary schools [87]. For PAL specifically, qualitative research has often reflected space concerns amongst teachers, suggesting the physical environment could be altered to support initiatives such as PAL, or that creative approaches are necessary to adapt PAL to work in available spaces [35,38,39,44,57,58,69,88,89]. Participants in the current study highlighted a preference for outdoor PAL delivery, which was also favoured in a Norwegian secondary school PAL program [38]. Time for planning and delivering PAL has been another key barrier highlighted in the literature [36,38,39,44,57,58,69,88]. This is consistent with

this study's qualitative findings. However the quantitative findings suggested teachers didn't believe PAL would be too time-consuming. Further, interviewed participants indicated that time concerns could be addressed by having the correct support in place, such as continuing professional development, access to subject-specific PAL resources, and a community of practice with other teachers and school leadership.

While state-funded schoolteachers expected administration support to impact PAL adoption, participants at independent schools did not. As a possible explanation, interviewed independent-school teachers described having more autonomy in implementing new approaches. Therefore, a particular focus on state-funded school settings could enable greater universality in the application of future PAL research. Previous qualitative studies concur with the necessity of suitable school support and consistency, to maintain PAL long-term through its implementation complexities [38,69,90], even when a relevant national policy exists [53].

Teacher recommendations for a whole-school approach align with a 2021 meta-synthesis of qualitative PAL research with UK primary school teachers [35] and the Creating Active Schools framework [33]. Collaboration between teachers, sharing resources and feedback within or across schools, was considered valuable to strong PAL implementation, here and in previous qualitative literature [38,49,90]. Cross-curricular collaboration was also welcomed by most participants, which Norwegian teachers previously also felt could be a potentially transformative pedagogical approach [36].

## Strengths and limitations

The mixed methods approach was a key strength, whereby triangulation between qualitative and quantitative components, with phenomena expanded and explained through different lenses, enhanced the depth of understanding [91]. The findings contribute new evidence to the field, addressing a scarcity of research on this topic at the secondary level, particularly in the UK [27]. The study was the first to stratify PAL data by subject and school type, providing important new insight into variation in acceptability and delivery across different educational settings. The inclusion of teachers from a range of roles, subjects, and school types in the qualitative component broadened the perspectives captured and enabled more nuanced insight, enhancing the transferability of findings. [66,92]. Another strength was the use of the framework method, which is suggested to augment the credibility and relevance of findings [65].

Notwithstanding these strengths, there are some limitations to the current study design. First, the TEDx video shown to participants, while intended to standardise understanding, may have introduced bias by framing PAL in a positive light. Second, adherence to this protocol was inconsistent: 42.7% of survey respondents did not watch the video, potentially limiting their understanding of PAL and introducing variability in how the concept was interpreted. Third, the recruitment strategy, using purposive, convenience, and snowball sampling, may have introduced self-selection bias. The approach may have attracted individuals already interested or engaged in PA in education, limiting the representativeness of the sample and potentially inflating the reported acceptability of PAL.

Fourth, the sample lacked demographic diversity. Interviewees were all white and non-disabled, and survey respondents were predominantly female and more likely to teach physically active subjects. These characteristics may have contributed to more favourable attitudes toward PAL and limit the transferability and generalisability of the findings.

Fifth, the sample size (N = 75) may have been under-powered to accurately identify differences. Sixth, survey items were adapted or developed for this study and did not undergo formal psychometric validation. While piloting in a small informal group supported face validity, this does not confirm that items accurately captured the intended constructs, and reliability of the measures remains uncertain. Future research should aim for more representative and large-scale studies with validated instruments.

## Implications and recommendations

The study findings regarding the acceptability and delivery of PAL generally align with previous research in other settings, whilst expanding the evidence base to consider a UK secondary school context. For the first time, differences in teacher

perspectives have been examined between gender, subject, and school type, with the results highlighting possible future focus areas such as younger years of secondary school, state-funded schools, and outdoor PAL. Certain considerations, such as lower acceptance amongst male teachers and humanities and social science teachers, and urban-located schools having particular space issues, should be further explored in future research.

Findings support a phased, subject-sensitive approach to PAL implementation at the school level, beginning with subjects where it is most feasible and acceptable, in order to generate early momentum. This strategy can help build confidence and demonstrate impact, while laying the groundwork for broader uptake across the curriculum. At the policy level, these insights could inform scalable models that encourage flexible and context-responsive adoption of PAL across diverse school settings [93].

Input from other key stakeholders is now necessary to continue assessing feasibility and acceptability, and eventually design appropriate interventions and subject-specific resources for UK secondary schools [45]. In particular, engagement with adolescents is essential, as the receivers of PAL [45]. A previous Norwegian study explored adolescents' experiences, and found classroom PA was meaningful to pupils, yet highlighted challenging implications for social dynamics [94]. Insight into UK-specific adolescent perspectives is necessitated. Special schools, for pupils with severe disabilities, may require a different approach with specific research, due to the pupils' unique needs [51].

In practice, while more evidence and resources are needed, there are already opportunities for teachers to add in more physically active tasks within curricular lessons. This could include running to corners of the room to answer questions, making a dance to interpret a poem, pupils acting as cells to illustrate respiration, taking the class outside, or collaborating with a drama, dance, or PE teacher. The indicated acceptability suggests more funding in this field is warranted, and future research progress may lead to eventual policy change to support and motivate PAL implementation.

## Conclusion

PAL has potential to address challenges encountered previously in implementing effective PA interventions in secondary schools, and to improve adolescent PA levels and academic attainment. This study found that teachers generally find the idea of PAL acceptable, and a potential means of updating an educational system perceived to be outdated. Concerns were raised regarding how realistic PAL delivery might be in a pressured environment, while possible challenges were anticipated in engaging girls and older year groups. However, teachers conveyed implementation could be feasible if suitable evidence, support, and resources were available through a whole-school approach. The findings indicate further research investigating PAL in UK secondary schools is warranted and highlight potential next steps and considerations.

## Supporting information

**S1 File. Interview topic guide.**
(PDF)

**S2 File. Analytical framework categories, and definitions, for qualitative analysis.**
(PDF)

**S3 File. Outline of the themes of PAL acceptability amongst UK secondary school teachers, with examples from the data.**
(PDF)

**S4 File. STROBE Statement—checklist of items that should be included in reports of observational studies.**
(PDF)

**S5 File. COREQ (COnsolidated criteria for REporting Qualitative research) checklist.**
(PDF)

**S1 Table. Data underlying the presented findings: Teachers' reported current classroom physical activity, with differences by participant gender, school location, and school type.** Abbreviation: IQR – interquartile range. Bold p-value numbers represent statistical significance of p < 0.05.
(XLSX)

**S2 Table. Data underlying the presented findings: PAL acceptability amongst teachers, with differences by participant gender, school location, and school type.** Abbreviation: IQR – interquartile range. PAL – physically active learning. Bold p-value numbers represent statistical significance of p < 0.05.
(XLSX)

**S3 Table. Data underlying the presented findings: Teacher responses for PAL delivery variables, with differences by gender, school location, and school type.** Abbreviation: IQR – interquartile range. PAL – physically active learning. Bold p-value numbers represent statistical significance of p < 0.05.
(XLSX)

**S4 Table. Additional underlying data for reported findings.**
(XLSX)

## Acknowledgments

The authors would like to thank everyone who participated in this research, and Dr Miranda Armstrong for her guidance and support.

## Author contributions

**Conceptualization:** Lara E Hollander, Lydia Emm-Collison.

**Data curation:** Lara E Hollander.

**Formal analysis:** Lara E Hollander.

**Investigation:** Lara E Hollander.

**Methodology:** Lara E Hollander, Lydia Emm-Collison.

**Project administration:** Lara E Hollander.

**Resources:** Lara E Hollander.

**Supervision:** Zoi Toumpakari, Lydia Emm-Collison.

**Visualization:** Lara E Hollander, Zoi Toumpakari, Lydia Emm-Collison.

**Writing – original draft:** Lara E Hollander.

**Writing – review & editing:** Lara E Hollander, Zoi Toumpakari, Lydia Emm-Collison.

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
