## [Decision Letter · Decision Letter 0]

12 May 2025

Dear Dr. Hollander,

Thank you for submitting your manuscript to PLOS ONE. After careful consideration, we feel that it has merit but does not fully meet PLOS ONE’s publication criteria as it currently stands. Therefore, we invite you to submit a revised version of the manuscript that addresses the points raised during the review process.

We look forward to receiving your revised manuscript.

Kind regards,

Jindong Chang, Ph.D.

Academic Editor

PLOS ONE

Journal Requirements:

2. In the online submission form, you indicated that [Due to ethical restrictions prohibiting public sharing of the data set, as participants did not agree to this, researchers may contact the lead author to request the data.].

Reviewers' comments:

Reviewer's Responses to Questions

**Comments to the Author**

1. Is the manuscript technically sound, and do the data support the conclusions?

Reviewer #1: Yes

Reviewer #2: Yes

2. Has the statistical analysis been performed appropriately and rigorously?

Reviewer #1: Yes

Reviewer #2: Yes

3. Have the authors made all data underlying the findings in their manuscript fully available?

Reviewer #1: Yes

Reviewer #2: No

4. Is the manuscript presented in an intelligible fashion and written in standard English?

Reviewer #1: Yes

Reviewer #2: Yes

Reviewer #1: Dear authors,

Firstly, I would like to express my gratitude for the opportunity to collaborate on a research project of significant importance and relevance, particularly in light of the challenges posed by traditional teaching methods in the face of a technological surge that has the potential to distract many children and adolescents. The study emphasises that physical activity and more active learning methods are not yet accorded sufficient priority by many schools, with the focus remaining on curricular performance. It is not that the importance of curricular performance is diminished, but rather that a more active teaching method can contribute to the integral development of the individual.

Consequently, I would like to offer some observations on subjects that could benefit from further elaboration.

On line 153, page 7, the following assertion is made: As this exploratory study sought to establish a foundation for future research, the recruitment target was set at 75-150 survey respondents, with a subsample of 7-10 interviewees, based on the findings of previous similar studies [38, 50, 51, 52, 53, 54]. However, the text does not specify the number of participating schools or if a minimum number of schools was required for recruitment.

It is to be posited herewith that, in the event of 25 teachers from each school participating, a total of 75 individuals would be required. Conversely, should only 8 teachers from 10 schools participate, the resultant sample would already comprise 80 individuals. The issue under discussion is that of the representativeness of the schools. It is posited that the inclusion of a greater number of teachers from a given unit could give rise to problems.

It is acknowledged that further information can be found in other sources. The integration of a sentence could facilitate comprehension.

Despite the implementation of a multifaceted sampling strategy, encompassing purposive, convenience, and snowball sampling techniques, this approach may potentially compromise the representativeness of the participant cohort. This particular recruitment strategy tends to appeal to teachers who have already demonstrated a high level of engagement or interest in the domain of PAL (Physical Activity in Learning), thereby introducing a form of self-selection bias. This may compromise the generalisability of the findings. Despite the recognition of such limitations in previous literature, it is imperative to articulate the notion of self-selection bias.

Finally, it is imperative to emphasise the absence of validation for the employed measures, a subject that demands further attention. It is important to note that some of the questions in the questionnaire were created by the researcher himself. These were based on previous literature, but without undergoing a formal psychometric validation process. Moreover, the preliminary study was confined to a sample size of four participants, comprising one teacher and three peers. While this approach offers face validity, there is no guarantee that the items accurately capture the proposed constructs, such as acceptance, motivation, or perception of PAL. The adaptation of existing scales (for example, the Work Tasks Motivation Scale) involves changes in structure and items without revalidation, which may affect reliability and construct validity.

Reviewer #2: Summary

Thank you for the opportunity to review this work. The study investigates teachers’ perspectives regarding the acceptability and feasibility of implementing Physically Active Learning (PAL) in UK secondary schools. A mixed methods design was employed, with quantitative survey data (N=75), qualitative survey data (N=63) and semi-structured interviews (N=7). The authors conclude that teachers were generally positive about the idea of introducing PAL into their teaching practice, although viewpoints differed across subject areas and between males and females. Potential implementation challenges are discussed.

This paper is an interesting read and has a lot of potential to contribute to the nascent body of research on this topic. I particularly commend the authors on their efforts to include a broad cross-section of teachers, representing different subject areas, school types and rural and urban settings, which enabled them to capture a wide range of viewpoints. My main points for consideration relate to aspects of the methods description, data presentation and clarity of the writing in some sections.

With respect to the journal policies, I note that there are restrictions on public data availability due to consent issues, although data may be made available on request.

Specific points to consider are provided below:

Methods

1. As highlighted in the COREQ checklist, the authors have not included researcher credentials, experience/training or key characteristics (COREQ items (2, 5 & 8). It seems that these points would be simple to rectify, and showing how these issues were considered and/or mitigated would add rigour to the study.

2. Lines 133-140 (study design): The purpose of utilizing a mixed methods design is well described. For even greater clarity, it may be helpful to explicitly state whether the intention was to give greater weight to qual data (which seems to be implied) or equal weight to both quan & qual data.

3. Line 144 (study design): “… combined at the final stage” - Please provide more detail on the integration. How were quan and qual data combined and what is meant by the “final stage”?

4. Lines 184-244 (survey): Consider whether this whole section could be restructured or more concisely summarised for greater clarity. E.g. line 195 mentions “Two measures adapted from previous studies …” but these measures are not named or referenced until subsequent paragraphs.

5. Lines 286-316 (qualitative analysis): This section could benefit from a clearer overall structure and more detail in some areas. Consider utilising the 7 step process (lines 305-313) to describe was done at each stage. The transcription process described in an earlier section (lines 271-273) may be more appropriate here under step 1.

6. Line 297: Here the socio-ecological model is mentioned. However, it is not apparent how (or if) the final themes were mapped onto this model. Some clarification is needed. Describing what was done at each of the seven stages may help to resolve this.

Results

7. Ensure numbers are consistently formatted. E.g. percentages are reported with a mix of 0 or 1 d.p. in Table 1 and in-text reporting. Similarly for p-values.

8. Figures 2, 3 and 6 are somewhat difficult to digest due to the number of colours/groups, questions, and pairwise comparisons. Consider whether this data could be presented more clearly in a table, which could allow for complete reporting of precise p-values for pairwise comparisons, both significant and non-significant.

9. Figure 5 is a little unclear as it uses tonally similar colours (blue and green). Consider using graded shades from dark to light to improve readability (considering that the article may sometimes be printed in black and white) or perhaps include this information concisely in the text without a figure.

Discussion

10. lines 604-607: The argument is a little unclear regarding a whole-school approach vs. targeted to subject areas. Could these two approaches fit together or are they competing ideas?

11. lines 677-682 (limitations): could be expressed more clearly. It seems two separate limitations are highlighted here – i) use of a pro-PAL TEDx talk – which may have introduced bias, and ii) lack of adherence to the protocol - almost half did not watch the video, which may have limited their understanding of the topic (as mentioned earlier in the article). It would be helpful to separate these two points. The suggestion that lack of adherence to watching the video was a bonus as it reduced bias would seem to undermine the protocol. Instead, it may be better to simply state its limitations.

12. lines 684-697 (limitations): similar to the above point, the limitations mentioned in this paragraph could also be more clearly delineated. Perhaps consider whether it could help to use a transition chain such as first, second, … to distinguish the points being addressed throughout the limitations section.

Supplementary materials

13. Ensure tables in supplementary materials have a header and any footnotes required to aid interpretation.

14. Ensure supplementary materials are referenced at relevant points throughout the text.

15. Line 418: (examples of quotes) references S1 file but seems to be referring to data in S3.

General points

16. A final grammar check should pick up any minor grammatical errors – e.g. take care with use of “however”, which should not be used to join two sentences (l 604 and elsewhere).

**Do you want your identity to be public for this peer review?** For information about this choice, including consent withdrawal, please see our Privacy Policy

Reviewer #1: **Yes: ** Magno Conceição Garcia

Reviewer #2: No

---

## [Author Response · Author response to Decision Letter 1]

24 Jun 2025

Dear Dr Chang,

Thank you for the opportunity to revise our manuscript and respond to the helpful and constructive feedback from you and the reviewers.

Please find below our point-by-point responses to all comments raised, with references to changes made in the revised manuscript. We believe the revisions have strengthened the manuscript, and we hope that it is now suitable for publication in PLOS ONE.

Editorial Requirements

1. PLOS ONE formatting style:

We have checked that the manuscript conforms to PLOS ONE formatting requirements, with relevant changes made, including file names, to conform to the provided templates.

2. Data availability:

We have updated the Data Availability Statement to clarify that a minimal dataset, in line with PLOS ONE’s data availability guidelines, has now been included as supporting information (S1-S4 Tables). These files provide summary-level data underpinning the quantitative findings, such as group-level frequencies, percentages, and statistical test results. In accordance with participant consent, where individuals were informed that no individual results would be published or made available, raw individual-level data are not shared.

3. Reference check:

We have reviewed the reference list. No retracted papers were cited. Minor reference formatting updates have been made where necessary, and these changes are reflected in the revised manuscript.

------------

Reviewer #1 Comments and Author Responses

We are very grateful for the positive comments and the detailed suggestions you have provided.

1. [On line 153, page 7, the following assertion is made: As this exploratory study sought to establish a foundation for future research, the recruitment target was set at 75-150 survey respondents, with a subsample of 7-10 interviewees, based on the findings of previous similar studies [38, 50, 51, 52, 53, 54]. However, the text does not specify the number of participating schools or if a minimum number of schools was required for recruitment.

It is to be posited herewith that, in the event of 25 teachers from each school participating, a total of 75 individuals would be required. Conversely, should only 8 teachers from 10 schools participate, the resultant sample would already comprise 80 individuals. The issue under discussion is that of the representativeness of the schools. It is posited that the inclusion of a greater number of teachers from a given unit could give rise to problems.

It is acknowledged that further information can be found in other sources. The integration of a sentence could facilitate comprehension.]

Thank you for raising this point regarding the representativeness of schools in the sample. As this study largely recruited teachers via social media and professional contacts, we anticipated that it would not be feasible to track the number of schools represented. Rather than focusing on a specific number of schools, our aim, in this preliminary stage of secondary school PAL research, was to ensure broad representation across school types, subject areas, and teacher roles. This was supported by demographic data collected through the survey, which enabled us to monitor diversity and attempt to adjust recruitment as needed. For example, by approaching networks serving underrepresented subjects or school contexts. We have now added a clarifying sentence in the Methods section (Page 7 Lines 157-161) to reflect this approach.

2. [Despite the implementation of a multifaceted sampling strategy, encompassing purposive, convenience, and snowball sampling techniques, this approach may potentially compromise the representativeness of the participant cohort. This particular recruitment strategy tends to appeal to teachers who have already demonstrated a high level of engagement or interest in the domain of PAL (Physical Activity in Learning), thereby introducing a form of self-selection bias. This may compromise the generalisability of the findings. Despite the recognition of such limitations in previous literature, it is imperative to articulate the notion of self-selection bias.]

The reviewer makes an important point here. We have consequently expanded the limitations section to explicitly discuss the potential for self-selection bias due to the sampling strategies (Page 32 Lines 692-696).

3. [Finally, it is imperative to emphasise the absence of validation for the employed measures, a subject that demands further attention. It is important to note that some of the questions in the questionnaire were created by the researcher himself. These were based on previous literature, but without undergoing a formal psychometric validation process. Moreover, the preliminary study was confined to a sample size of four participants, comprising one teacher and three peers. While this approach offers face validity, there is no guarantee that the items accurately capture the proposed constructs, such as acceptance, motivation, or perception of PAL. The adaptation of existing scales (for example, the Work Tasks Motivation Scale) involves changes in structure and items without revalidation, which may affect reliability and construct validity.]

We have revised the limitations section to clarify that survey items were newly developed or adapted without formal psychometric testing, emphasising the subsequent uncertainties around construct validity and reliability. (Pages 32-33 Lines 705-708)

---------------

Reviewer #2 Comments and Author Responses

We are very grateful for the positive comments and the detailed suggestions you have provided.

Methods

1. [As highlighted in the COREQ checklist, the authors have not included researcher credentials, experience/training or key characteristics (COREQ items (2, 5 & 8). It seems that these points would be simple to rectify, and showing how these issues were considered and/or mitigated would add rigour to the study.]

We have added a line detailing the lead researcher’s characteristics, training, relevant experience, and reflexivity practices on Page 11 Lines 240-244.

2. [Lines 133-140 (study design): The purpose of utilizing a mixed methods design is well described. For even greater clarity, it may be helpful to explicitly state whether the intention was to give greater weight to qual data (which seems to be implied) or equal weight to both quan & qual data.]

We now explicitly state the intention was to give equal weight to qual and quant data in the mixed methods design (Page 6 Line 137).

3. [Line 144 (study design): “… combined at the final stage” - Please provide more detail on the integration. How were quan and qual data combined and what is meant by the “final stage”?]

We have revised the sentence to explain the process more clearly, and removed the wording “final stage”, instead describing the specific point in the process when integration occurred (i.e. following the separate analysis of quant and qual data) (Page 7 Lines 144-147).

4. [Lines 184-244 (survey): Consider whether this whole section could be restructured or more concisely summarised for greater clarity. E.g. line 195 mentions “Two measures adapted from previous studies …” but these measures are not named or referenced until subsequent paragraphs.]

Thank you to the reviewer for highlighting the need for greater clarity. We restructured the entire section for a clearer, more logical flow. (Pages 9-10 Lines 192-231)

5. [Lines 286-316 (qualitative analysis): This section could benefit from a clearer overall structure and more detail in some areas. Consider utilising the 7 step process (lines 305-313) to describe was done at each stage. The transcription process described in an earlier section (lines 271-273) may be more appropriate here under step 1.]

We have expanded the qualitative analysis section using the 7-step framework method process, to enhance the structure and level of detail. We have also integrated the transcription process into this section (Pages 13-14 Lines 290-314)

6. [Line 297: Here the socio-ecological model is mentioned. However, it is not apparent how (or if) the final themes were mapped onto this model. Some clarification is needed. Describing what was done at each of the seven stages may help to resolve this.]

Using the seven stages, we have clarified the role of the socio-ecological model within the qualitative analysis, and how it served as a structure to assist thematic interpretation through its predominant role in the analytical framework generated (rather than mapping the final themes onto this model). (Page 13 Lines 300-310). A reference to the relevant Supplementary File (S2 File) has also been added to help the reader more clearly understand the use of the model in the framework.

Results

7. [Ensure numbers are consistently formatted. E.g. percentages are reported with a mix of 0 or 1 d.p. in Table 1 and in-text reporting. Similarly for p-values.]

All percentages are now reported to 1 decimal place. P-values follow PLOS ONE’s guidance (exact values reported when ≥ 0.001).

8. [Figures 2, 3 and 6 are somewhat difficult to digest due to the number of colours/groups, questions, and pairwise comparisons. Consider whether this data could be presented more clearly in a table, which could allow for complete reporting of precise p-values for pairwise comparisons, both significant and non-significant. ]

Thank you for this feedback. In drafts, these were initially presented as tables. However, they looked too unwieldly; we found that the figures were clearer to digest. We have considered the reviewer’s concern and opted to retain the figures for a visual summary, while adding supplemental tables showing full pairwise comparisons with exact p-values, and more detailed summaries of the findings and statistical tests. (See S1-S4 Tables)

9. [Figure 5 is a little unclear as it uses tonally similar colours (blue and green). Consider using graded shades from dark to light to improve readability (considering that the article may sometimes be printed in black and white) or perhaps include this information concisely in the text without a figure.]

We have removed the figure and integrated its content into the text instead, as suggested (Page 17 Lines 371-373).

Discussion

10. [lines 604-607: The argument is a little unclear regarding a whole-school approach vs. targeted to subject areas. Could these two approaches fit together or are they competing ideas?]

The reviewer makes a very interesting point. We have clarified how these are not competing approaches, but could function as complementary strategies that can be phased over time, starting with the structured targeting of subjects that hold higher acceptability and feasibility, to then inform wider integration across the school. This is now described on Pages 28-29, Lines 607-616, and in the Implications and Recommendations section on Page 33 Lines 722-728.

11. [lines 677-682 (limitations): could be expressed more clearly. It seems two separate limitations are highlighted here – i) use of a pro-PAL TEDx talk – which may have introduced bias, and ii) lack of adherence to the protocol - almost half did not watch the video, which may have limited their understanding of the topic (as mentioned earlier in the article). It would be helpful to separate these two points. The suggestion that lack of adherence to watching the video was a bonus as it reduced bias would seem to undermine the protocol. Instead, it may be better to simply state its limitations.]

As suggested, we have separated the TEDx video issues into two limitations: (i) bias introduced by the video, and (ii) lack of adherence to the video protocol. We have also removed the previous suggestion that non-adherence reduced bias, and instead focus solely on the limitations. (Page 32 Lines 688-692)

12. [lines 684-697 (limitations): similar to the above point, the limitations mentioned in this paragraph could also be more clearly delineated. Perhaps consider whether it could help to use a transition chain such as first, second, … to distinguish the points being addressed throughout the limitations section.]

We have revised the structure using “First,” “Second,” etc. to enhance readability. (Page 32 Lines 688-705)

Supplementary materials

13. [Ensure tables in supplementary materials have a header and any footnotes required to aid interpretation.]

Headers have been added to supplementary files, with footnotes added where appropriate.

14. [Ensure supplementary materials are referenced at relevant points throughout the text.]

Supplementary materials are now referenced at appropriate points in the text, including an S1 File reference on Page 11 Line 251, and S2 File reference on Page 13 Line 304.

15. [Line 418: (examples of quotes) references S1 file but seems to be referring to data in S3.]

The incorrect reference to S1 has been corrected to S3. (Page 20 Line 421)

General points

16. [A final grammar check should pick up any minor grammatical errors – e.g. take care with use of “however”, which should not be used to join two sentences (l 604 and elsewhere).]

We have completed a final grammar and style check, including appropriate use of words such as “however.”

---

## [Editor Report · Decision Letter 1]

1 Jul 2025

Teacher acceptability of physically active learning in UK secondary schools – a mixed methods study

PONE-D-25-14294R1

Dear Dr. Hollander

We’re pleased to inform you that your manuscript has been judged scientifically suitable for publication and will be formally accepted for publication once it meets all outstanding technical requirements.

Kind regards,

Jindong Chang, Ph.D.

Academic Editor

PLOS ONE
---

## [Editor Report · Acceptance letter]

PONE-D-25-14294R1

PLOS ONE

Dear Dr. Hollander,

I'm pleased to inform you that your manuscript has been deemed suitable for publication in PLOS ONE. Congratulations! Your manuscript is now being handed over to our production team.

Kind regards,

on behalf of

Dr. Jindong Chang

Academic Editor

PLOS ONE